# Human Pancreatic Islets React to Glucolipotoxicity by Secreting Pyruvate and Citrate

**DOI:** 10.3390/nu15224791

**Published:** 2023-11-15

**Authors:** Johan Perrier, Margaux Nawrot, Anne-Marie Madec, Karim Chikh, Marie-Agnès Chauvin, Christian Damblon, Julia Sabatier, Charles H. Thivolet, Jennifer Rieusset, Gilles J. P. Rautureau, Baptiste Panthu

**Affiliations:** 1Laboratoire CarMeN, UMR INSERM U1060/INRAE U1397, University of Lyon, Université Claude Bernard Lyon 1, 69310 Pierre-Bénite, France; 2Department of Endocrinology and Diabetes, Hospices Civils de Lyon, Hopital Lyon Sud, 69310 Pierre-Bénite, France; 3Unité de Recherche MolSys, Faculté des Sciences, Université de Liège, 99131 Liège, Belgium; 4Laboratory of Cell Therapy for Diabetes (LTCD), PRIMS Facility, Institute for Regenerative Medicine and Biotherapy (IRMB), University Hospital of Montpellier, 34295 Montpellier, France; 5Centre de Résonance Magnétique Nucléaire à Très Hauts Champs, UMR 5082 CNRS, ENS Lyon, UCBL, Université de Lyon, 69100 Villeurbanne, France

**Keywords:** human pancreatic islets, INS-1E, NMR, glucolipotoxicity, GSIS, insulin content, quantitative metabolomics, type 2 diabetes

## Abstract

Progressive decline in pancreatic beta-cell function is central to the pathogenesis of type 2 diabetes (T2D). Here, we explore the relationship between the beta cell and its nutritional environment, asking how an excess of energy substrate leads to altered energy production and subsequent insulin secretion. Alterations in intracellular metabolic homeostasis are key markers of islets with T2D, but changes in cellular metabolite exchanges with their environment remain unknown. We answered this question using nuclear magnetic resonance-based quantitative metabolomics and evaluated the consumption or secretion of 31 extracellular metabolites from healthy and T2D human islets. Islets were also cultured under high levels of glucose and/or palmitate to induce gluco-, lipo-, and glucolipotoxicity. Biochemical analyses revealed drastic alterations in the pyruvate and citrate pathways, which appear to be associated with mitochondrial oxoglutarate dehydrogenase (OGDH) downregulation. We repeated these manipulations on the rat insulinoma-derived beta-pancreatic cell line (INS-1E). Our results highlight an OGDH downregulation with a clear effect on the pyruvate and citrate pathways. However, citrate is directed to lipogenesis in the INS-1E cells instead of being secreted as in human islets. Our results demonstrate the ability of metabolomic approaches performed on culture media to easily discriminate T2D from healthy and functional islets.

## 1. Introduction

Pancreatic beta-cell islets secrete insulin in response to glucose stimulation. Chronic exposure of human beta cells and islets to supraphysiological concentrations of glucose, known as glucotoxicity, impairs their secretion of glucose-stimulated insulin (GSIS) [1,2]. In addition, numerous studies have shown the negative impact of high concentrations of free fatty acids, known as lipotoxicity, on beta-cell function and mass [3]. Overall, physiological gluco- or lipotoxicity leads to type 2 diabetes (T2D), of which beta-cell dysfunction is considered a hallmark [4,5]. The mechanisms behind this loss of function remain unknown. However, mitochondria appear to play a key role, as it has been shown that increasing the ATP/ADP ratio leads to exocytosis of insulin granules. The increase in the ATP/ADP ratio during glucose stimulation is linked to a close link between glycolysis and mitochondrial metabolism [6]. Glucokinase expression and low levels of lactate dehydrogenase enable beta cells to efficiently convert glucose into pyruvate, which is incorporated into the tricarboxylic acid (TCA) cycle and oxidative phosphorylation to increase ATP production [7,8,9,10]. Chronic exposure to high glucose concentration reduces beta-cell ATP production, altering insulin content (IC) and secretion in pancreatic islets and the beta-cell line [11,12,13].

The question of how the glycolysis–TCA–insulin secretion axis is altered in T2D, or during chronic glucolipotoxicity under in vitro conditions, remains unresolved [14]. Multi-omic analyses were performed to answer these questions. Intracellular metabolomic analyses carried out on beta cells during glucotoxicity and/or lipotoxicity treatments revealed mitochondrial metabolic disturbances [15,16]. When chronically exposed to high concentrations of glucose, murine-beta-cell lines showed a rerouting of glucose to glycerol and an increase in lipid production in rodents [17]. Lastly, transcriptomic and proteomic analyses were obtained from islets isolated from drug-induced diabetic mice, showing altered mitochondrial metabolism. This deficiency is associated with a decrease in mitochondrial TCA enzymes, while enzymes associated with glycolysis are upregulated [18]. Independently, phosphoproteomic analyses performed on islets isolated from obese *db*/*db* mice showed specific inhibition of pyruvate dehydrogenase by phosphorylation and reduction of the two mitochondrial dehydrogenases involved in the TCA cycle, namely, isocitrate dehydrogenase (IDH) and oxoglutarate dehydrogenase (OGDH) [19]. Finally, a recent review of metabolic studies points out that most metabolomic analyses have focused on the intracellular metabolome, but never on the consumption or secretion of extracellular metabolites in the islet environment. Moreover, only a few studies have been carried out on islets isolated from humans, and most of these have used rodent cell lines and islets [20].

In this study, we used proton nuclear magnetic resonance (NMR) metabolomic approaches to investigate metabolic alterations in human islets and the INS-1E cell line. Human islets from non-diabetic and diabetic donors and the murine-beta-cell line INS-1E were cultured under conditions of high glucose exposure, to simulate glucotoxicity, high palmitate exposure, to simulate lipotoxicity, and high glucose and palmitate exposure, to simulate glucolipotoxicity. We highlight the functional and metabolic alterations of both models in the context of gluco-, lipo-, and glucolipotoxicity. Our data confirm the alterations in mitochondrial metabolism and show that mitochondrial pyruvate and citrate dehydrogenases are altered checkpoints under conditions of glucolipotoxicity. Interestingly, such metabolic alterations make it easy to distinguish functional from non-functional islets.

## 2. Materials and Methods

### 2.1. Ethic Statement

Human islets were isolated from the pancreas of 9 non-diabetic and 2 diabetic multiple organ donors (Table 1). The study was conducted in accordance with the Declaration of Helsinki, and the protocol was approved by the ethics committee of the University of Lyon. Human pancreases were harvested from brain-dead organ donors after obtaining written informed consent from donor (HI-28) or family members. Isolation of the human islets was performed by the European diabetes study center (CEED, Strasbourg, France), the Geneva university hospitals (ECIT collaboration, Geneva, Switzerland), and the laboratory of cellular therapy for diabetes (LCTD, Montpellier, France, Biological Resources center, Collection 5 “Human Islets of Langerhans”, 345 identifiant Biobanque n° BB-0033-00031/PFS13-008).

### 2.2. Human Islet Isolation and Culture

Human islets were isolated by enzymatic digestion and density gradient purification from the pancreas of multiorgan donors, as described elsewhere [21]. The isolated islets were then cultured in DMEM medium with 5.5 mmol/L glucose until the experiments were carried out. Photographs of the islets were taken using the ZEN microscope version 2012 (Carl Zeiss Microscopy GmbH, France).

### 2.3. Cell Culture

Human islets were cultured for 1 day in DMEM (Dutcher L0064-500) supplemented with 5 mM glucose, 10% FBS, 2 mM glutamine, 100 U/mL penicillin, and 100 g/mL streptomycin. For specific metabolomic culture conditions, the medium was replaced with DMEM without phenol red (Pan Biotech P04-01548, France), and supplemented with 10% FBS, 4 mM glutamine 100 U/mL penicillin, and 100 g/mL streptomycin, 5 or 25 mM glucose (Sigma-Aldrich G8270-100G, France) and 0.017 mM BSA (Roche diagnostic 10 775 835 001, France) or 0.1 mM palmitate: BSA (6:1) (Sigma-Aldrich P0500-10G, France). All the data presented were obtained after 48 h of metabolic stress condition, as this duration was chosen to be both minimal and sufficient to induce detectable modifications of islet physiology, as previously described [22].

INS-1E beta-cells (RRID:CVCL_0351) were grown in a humidified atmosphere (5% CO_2_, 95% air at 37 °C) in RPMI (Dutcher L0501-500, France) supplemented with 10 mM glucose, 5% FBS, 4 mM pyruvate, 4 mM glutamine, 50 µM β-mercaptoethanol, 100 U/mL penicillin, and 100 g/mL streptomycin. For specific metabolomic culture conditions, the medium was replaced by DMEM without phenol red and supplemented with 5% FBS, 4 mM glutamine 100 U/mL penicillin, and 100 g/mL streptomycin, 10 or 25 mM glucose, and 0.017 mM BSA or 0.1 mM palmitate: BSA (6:1).

### 2.4. Insulin Assay

Human Insulin-Specific RIA was carried out at the Lyon-Sud hospital (radioimmunoassay, CBS, Lyon, France). Fifty human islets were preincubated 2 h at 37 °C in DMEM 5 mM glucose containing 5% FBS, 2 mM glutamine, 100 U/mL penicillin, and 100 g/mL streptomycin. Next, human islets were incubated for 45 min with 5.5 mM glucose in the same medium and challenged for 45 min with 25 mM glucose to measure the GSIS capacity of each batch of islets. Supernatants were collected after both 45 min incubation periods. Cellular insulin content was extracted by an overnight incubation in ethanol acid on 50 equivalent islets (IEQ). The samples were analyzed on the Cobas e411 and on the AutoDELFIA (CBS, Lyon, France). For the Cobas e411 analysis of insulin, the following reagents were used: Insulin CalSet, PreciControl Multimarker, PreciControl Multianalyte, PreciControl in the Centre de Biologie et de Pathologie Sud. A detection interval of 0.200–1000 μU/mL or 1.39–6945 pmol/L was defined by the manufacturer (COBAS Kit-insert Insulin, Roche Diagnostics, 2011). The assays used for the measurements on AutoDELFIA were B080–101 with a detection limit of 3 pmol/L and measurement interval 3–1000 pmol/L (AutoDELFIA Insulin Kit insert B080-101, PerkinElmer Life and Analytical Sciences, Turku, Finland, 2003). All measures were normalized on islets number.

INS-1E insulin assay was performed by ELISA (ALPCO 80-INSMS-E01) according to the manufacturer’s instructions and normalized on cell number. Briefly, cells were preincubated for 2 h at 37 °C in Krebs–Ringer bicarbonate HEPES buffer (NaCl 125 mM; KCl 4.74 mM; NaHCO_3_ 5 mM; CaCl_2_ 1 mM; MgSO_4_ 1.2 mM; K_2_HPO_4_ 1.2 mM; HEPES 25 mM; pH 7.4) containing 0.1% BSA and 5 mM glucose. For GSIS experiments, cells were incubated for 45 min in the same medium and then challenged for 45 min with 30 mM glucose in the same medium. Supernatants were collected after both 45 min incubation periods. Cellular insulin content was determined on overnight acid–ethanol-extracted lysates.

### 2.5. Triglyceride Assay

Triglyceride content and secretion were quantified according to the manufacturer’s instruction (Biolabo 87319, France) using glycerol from Sigma-Aldrich as standard and then normalized by the cell number.

### 2.6. Western Blot

Cell extracts from approximately 3000 Human islets and 10^7^ INS-1E cells were migrated on a 10% SDS-PAGE and transferred to PVDF membrane using a Bio-Rad Trans-Blot Turbo Transfer System and blotted using anti-tubulin (Sigma-Aldrich T5168), anti-ACC (Cell Signaling, France, #3676), ACLY (Cell Signaling #4332), Calnexin (Santacruz, France sc-6465), FASN (Cell Signaling #3180), IDH2 (Elabscience E-AB-11319, France), OGDH (Cell signaling #26865), PDH (Abclonal P08559, France), pPDH (Cell signaling #31866). All antibodies were diluted in powdered milk at a ratio 1/1000. The proteins of interest were revealed by enhanced chemiluminescence detection (Immobilon Forte Western HRP Substrate with a Bio-Rad Molecular Imager ChemiDoc XRS+, Universal Hood II, France) and the signal was quantified by using the Image Lab 6.1 software (Bio-rad, France).

### 2.7. Oil Red O Staining

INS-1E cells were washed twice with phosphate-buffered saline (PBS) and fixed using 3.7% formaldehyde (in PBS) for 1 h at room temperature. The fixative was removed and replaced with 60% isopropanol. Cells were stained for 10 min at room temperature with Oil Red O ready used solution (Merck, Darmstadt, Germany). Cells were then washed 3 times with PBS to remove excess stain. Photos were taken using ZEN Microscope version 2012 (Carl Zeiss Microscopy GmbH, France).

### 2.8. Oxygen Consumption Rate

INS-1E permeabilized cells were pelleted and resuspended in 100 µL KRBH supplemented with 0.5% free fatty acid BSA. Oxygen consumption rate (OCR) was measured at 37 °C using a Clark-type electrode (Mitocell MT200 with a 782 Oxygen meter, Strathkelvin Instruments, Scotland). Digitonin (50 µM) was added for 10 min to permeabilize cells. Pyruvate (2 mM) and ADP (2 mM) were used as respiratory substrates and were added during OCR acquisition.

### 2.9. Sample Preparation for NMR Analyses

Metabolite concentrations were measured in supernatants of INS-1E cells or human islets as previously described [23]. Cells were seeded at a density of 0.3 × 10^6^, 0.32 × 10^6^, 0.38 × 10^6^, or 0.42 × 10^6^ cells per well for control, glucotoxicity, lipotoxicity, or glucolipotoxicity conditions, respectively, in order to obtain the same number of cells upon supernatant collection (48 h after medium exchange, see Appendix A). Cells were cultured in 12-well dishes with 2 mL of culture medium. After 60 h, the medium was replaced by 600 μL of conditioned medium, and the culture was prolonged for 24 or 48 h as indicated. Control media without cells were processed along to determine the initial metabolite concentrations. At the end of the culture, supernatants were collected after centrifugation, snap-frozen, and stored at −20 °C until NMR analyses. For the NMR samples preparation, 540 μL of culture supernatants were supplemented with 60 μL of 10× phosphate buffer pH 7.4 (162 mM Na_2_HPO_4_, 34 mM NaH_2_PO_4_, 1 mM TSP, and 2 mM NaN_3_ in 100% D_2_O) [24] and vortexed. Samples were analyzed in 5 mm NMR tubes containing 550 μL of the sample mix. Control culture media were analyzed in parallel during each NMR session. For endometabolome analyses, we used the same optimized protocol as previously described [25,26]. Briefly, metabolite extraction was performed using 100% methanol on 2 × 10^7^ INS-1E cells. After extraction, raw metabolite extracts were dried under a gentle N2 flow until complete evaporation, then stored at −20 °C until NMR sample preparation, right before analysis. Then, 600 μL of 1× D_2_O phosphate buffer was used to redissolve dried extracts by vortex for 30 s. Extracts were then transferred to 1.5 mL Eppendorf tubes and centrifuged at 13,000 rpm for 1 min at 4 °C. Finally, 550 μL of supernatant was transferred to 5 mm NMR tubes.

### 2.10. NMR Analysis and Data Processing

High-resolution NMR analyses were carried out at the very high field NMR center (CRMN, Lyon, France). All NMR experiments were acquired at 30.0 °C on a 600 MHz Bruker NMR spectrometer (Bruker GmbH, Rheinstetten, Germany) equipped with a 5 mm TCI cryoprobe. A cooled SampleJet autosampler enabled high throughput data acquisition. A standard 1H-1D NMR pulse sequence nuclear Overhauser effect spectroscopy (NOESY) with z-gradient and water presaturation (Bruker pulse program noesygppr1d) was recorded on each sample, with a total of 128 transient free induction decays (FID) and a spectral width of 20 ppm, and a relaxation delay was set to 4 s. The NOESY mixing time was set to 10 milliseconds, and the 90° pulse length was automatically determined for each sample (around 13 μs). The total acquisition time of each sample was 12 min and 15 s.

All free induction decays (FIDs) were multiplied by an exponential function corresponding to a 0.3 Hz line-broadening factor prior to Fourier transform 1H-NMR spectra, which were manually phased and referenced to the glucose doublet at 5.23 ppm using TopSpin 3.6 (Bruker GmbH, Rheinstetten, Germany).

### 2.11. Metabolite Identification and Quantification

Identification of the metabolites was carried out from the 1D NMR spectra using the software Chenomx NMR Suite 8.0 (Chenomx Inc., Edmonton, AB, Canada) and confirmed from analysis of 2D 1H-1H TOCSY, 1H-13C HSQC, and 1H J-Resolved NMR spectra recorded with standard parameters. The measured chemical shifts were compared to reference shifts of pure compounds using the HMDB database [27]. Absolute metabolite concentrations were determined using Chenomx software by manual fitting of the proton resonance lines for the compounds available in the database. The linewidth used in the reference database was adjusted to the width of one component of the alanine doublet. A pure standard lactate solution (1 g/L, Fisher) was used as an external concentration reference and exploited using the ERETIC2 utility from TopSpin (Bruker GmbH, Rheinstetten, Germany) to add a digitally synthesized peak to a spectrum [28]. Concentrations are presented as non-normalized data. They are apparent concentrations, as endogenous and FCS proteins were not removed from the NMR samples.

### 2.12. Multivariate Analysis on Exometabolome

Outlier’s spectrum was identified by PCA on metabolite quantifications. The between-group differences were identified by OPLSDA coefficients on metabolite quantifications, and the implication of specific metabolite was assessed using VIP.

The quantified metabolites matrix was then introduced to Simca 15.0 software (https://www.sartorius.com/). Principal component analysis (PCA) and orthogonal partial least-squared discriminant analysis (OPLS-DA) were performed. All variables were Pareto scaled before analysis. Biomarkers were searched for using the Variable Importance in the Projection (VIP) on supervised OPLS-DA analysis technique between treatments. R^2^ and Q^2^ values were used to assess the quality of these models. The goodness of fit displaying the variance explained in the model and the predictability displaying the variance in the data predictable by the model [29] were indicated, respectively, by R^2^ and Q^2^. These values have to be as close as possible to 1.0 to indicate a good prediction for the model.

### 2.13. Correlogram and Heat Map

Correlation between beta-cell function, namely, glucose-stimulated insulin secretion (GSIS) and insulin content (IC) and exometabolome, were assessed after 48 h treatment. The r correlation coefficients were calculated using Excel. Statistical significance of correlation coefficients was assessed by using Student law. Heatmaps were generated using log2 (correlation coefficient fold change).

### 2.14. Statistical Analysis

All data are presented as mean ± SEM. If values are paired, differences between the two groups were tested with paired Student *t* test. Otherwise, value normality is assessed with Jarqe–Bera test, and differences between groups were tested with Welch test. Statistical significance was accepted when *p* < 0.05.

## 3. Results

### 3.1. Glucolipotoxicity-Induced Increased Glucose Intake Promotes Loss of Function of Glucose-Stimulated Insulin Secretion and Secretome Remodeling in Human Islets

To investigate the metabolic features associated with insulin secretion in beta cells, we studied cultures of human pancreatic islets isolated from eleven donors (from D1 to D11) (Appendix A; Table 1), among which two were isolated from donors known to have T2D (D10 and D11). We conducted Glucose-Stimulated Insulin Secretion (GSIS) measurements to assess beta-cell function after isolating and culturing the donor islets in a control medium containing 5 mM glucose. Dysfunctional islets were identified based on the absence of a significant insulin secretion response to acute glucose stimulation, as illustrated in Figure 1c. We further compared the mean basal insulin secretion for each donor to that of the GSIS-responding islet group, consisting of D1 to D5. Seven islet batches showed basal insulin secretion below 1000 mU/L/50 islets/45 min (D1–D7), and four others showed higher secretion levels (D8–D11) (Appendix A). As expected, lots D10 and D11 from T2D donors did not react to the GSIS test (Figure 1a and Appendix A). We observed that after 48 h of culture in the presence of 25 mM glucose and 0.1 mM palmitate, GSIS failed and IC decreased by more than 50%, confirming glucolipotoxicity-induced beta-cell dysfunction. Glucotoxicity alone, but not lipotoxicity alone, produced GSIS failure and strong IC decrease (Figure 1a,b and Appendix A; *p* < 0.05). To study the metabolic profiles of islets in relation to their GSIS properties and their ability to adapt to chronic nutritional stress, islet batches were divided (300 islets per tube) and subjected to modified culture conditions (six replicates per condition). After 48 h of culture, the supernatant was collected for nuclear magnetic resonance (1H-NMR) analysis. Sextuplicate of each batch was required to obtain robust quantification of metabolite consumption or production, independently of the natural variation in islet size for a given donor, as illustrated in Appendix A.

All cell culture supernatants provided high-resolution NMR spectra with sharp peaks typical of small molecules. Careful analysis of ^1^H 1D and the two-dimensional ^1^H-^1^H and ^1^H-^13^C NMR spectra identified and quantified 31 metabolites in each islet batch, under control vs. gluco-, lipo-, and glucolipotoxicity conditions (Appendix A).

Multivariate data analyses were first performed to determine metabolic secretome differences between healthy and T2D-derived islets under control culture conditions. A principal component analysis (PCA) was first performed to check the homogeneity of the samples. No biological or technical outliers were identified. PCA was not sufficient to discriminate between islets according to their insulin secretion capacity in response to GSIS (Appendix A). To optimize discrimination between islet metabolic profiles and identify functional metabolic signatures, a supervised analysis of NMR metabolic profiles was performed by orthogonal partial least-squared discriminant analysis (OPLS-DA) [30]. We obtained clear discrimination between a positive or negative GSIS response, as illustrated in Figure 1c and Appendix A, assessed by high values of goodness-of-fit model parameters R^2^ and Q^2^. These two components refer respectively to the variance explained and the variance predicted in the model (R^2^(X) = 0.489; R^2^(Y) = 0.94; Q^2^ = 0.914). The discrimination robustness was further validated by resampling the model 1000 times under the null hypothesis, which showed a clear decrease in R^2^ and Q^2^ with the correlation between the original and permuted class information Y matrices. The statistical significance of the model was assessed by a *p*-value of 9.25 × 10^−21^, calculated by analysis of variance (CV- ANOVA). Of the 31 metabolites quantified, 4 metabolites (citrate, alanine, glutamate, and lactate) showed a significant contribution to the statistical model, as shown by values of Variable Importance in Projection of independent variables (VIP) values greater than 1.5. Multivariate O-PLS-DA analyses were also performed under glucolipotoxicity conditions and showed equivalent discrimination of positive and negative islets (R^2^(X) = 0.682; R^2^(Y) = 0.923; Q^2^ = 0.871) (Appendix A). Alanine, pyruvate, and citrate appeared to have a significant influence on model discrimination, while glutamate and lactate are slightly below the VIP = 1.5 threshold. Altogether, alanine, pyruvate, and citrate emerged as biomarkers of human islet dysfunction, under both control conditions and glucolipotoxicity (Appendix A).

The impact of gluco-, lipo- and glucolipotoxicity conditions could only be assessed from D1 to D5 donors (Table 1). All 3 conditions induced significant alterations in metabolite secretion, which could be distinguished from the control by O-PLS-DA analysis (Figure 1d). Interestingly, glucolipo- (Appendix A) and glucotoxicity (Appendix A) induced similar disturbances, with pyruvate, citrate, alanine, and lactate being the most discriminating metabolites, while lipotoxicity alone impacted the secretion of a different set of metabolites (isoleucine, tryptophan, glutamate) (Appendix A).

Appendix A shows the absolute number of metabolites in nanomoles in the medium before islet incubation and after 48 h of exposure, allowing us to determine the rate of metabolite consumption or secretion in nmol per 300 islets per 48 h. Glucose was the most consumed metabolite with 271 (+/−107) nmol for healthy donors and up to 839 (+/−57) nmol (D10) consumed per 300 islets/48 h. No correlation could be established for glucose consumption under normal conditions between healthy islets and islets derived from donors with T2D. The same observation was made under glucolipotoxicity conditions, with glucose consumption unable to distinguish between positive and negative islets. Islet glucose consumption was increased upon glucolipotoxicity by 1.5 (D4) and up to 5-fold (D3). These results confirm that human islets do not regulate glucose entry under excess conditions in line with other reports in the literature [31]. Although lactate has been identified as a biomarker for discriminating islet-positive from islet-negative, its secretion varied from 55 (+/−30) nmol (D5) to 984 (+/−25) nmol (D10) with sometimes similar secretion for healthy islets D3 (680 (+/−63) nmol) and T2D islets D11 (695 (+/−123) nmol). The lactate secretion ratio between normal and glucolipotoxicity oscillates between 1.3 (D3, D10, D11) and 2.5 (D4) with no clear link to islet functionality.

Altogether, the variability of lactate secretion precludes its utility as a candidate biomarker of islet dysfunction.

### 3.2. Pyruvate and Citrate Secretion Are Biomarkers of Healthy Islets Alteration in the Glycolysis–TCA–Insulin Secretion Pathways

A correlogram was performed to compare GSIS, IC, and secretome metabolite characteristics only from D1 to D5 (Figure 1e). As expected, IC and GSIS were highly correlated, indicating that high insulin storage is associated with insulin secretory capacity. Alanine, citrate, and pyruvate secretions were associated with both IC and GSIS (*p* < 0.001). In healthy islets, we observed very low citrate secretion (≤10 nmol) in control conditions, while pyruvate secretion reached 30 nmol (Appendix A). Pyruvate and citrate secretions were significantly increased in T2D islets (D10 and D11; *p* < 0.01) (Figure 1f,g). Under glucolipotoxic conditions, pyruvate secretion reached more than 100 nmol and citrate secretion more than 20 nmol (Appendix A; Figure 1f,g). Statistical analyses comparing healthy and T2D-derived islets confirmed a significant increase in pyruvate (*p* < 0.01) and citrate secretion (*p* < 0.01). We conclude that culture conditions, particularly under high glucose concentration, may exacerbate a well-observed citrate and pyruvate over-secretion for T2D islets. This observation is easily confirmed by the plot of citrate secretion vs. function of pyruvate secretion, which shows a clear linear transition from islets positive in glucolipotoxicity to islets negative under normal conditions (Figure 1h). Taken together, these data indicate that citrate and pyruvate secretion is a quantitative biomarker of islet alteration and loss of insulin secretion under glucolipotoxic conditions.

Previous proteomic and phosphoproteomic studies have highlighted specific changes in mitochondrial enzymes associated with the diabetic mouse model [18,19] but natural variation in key metabolic enzymes is well observed between rodent and human islets [32]. Thus, we focused on mitochondrial dehydrogenases (pyruvate dehydrogenase (PDH) and its phosphorylated form (pPDH), isocitrate dehydrogenase 2 (IDH2) and mitochondrial oxoglutarate dehydrogenase (OGDH)). Quantitative immunoblots of the three dehydrogenases showed a decrease in OGDH expression of around 50% after exposure to glucotoxicity for four islets (D1, D5, D6, and D7) and no decrease for a single islet (D4), producing a significant overall decrease in OGDH (*p* < 0.09). Expression of PDH and IDH2 remained unchanged, as did the pPDH/PDH ratio (Figure 1i and Appendix A). Moreover, mitochondrial dehydrogenase expression was not clearly affected in T2D islets (Appendix A).

### 3.3. Rodent INS-1E Beta Cells Confirm and Modulate Glucolipotoxicity-Related Stopper in Glycolysis–TCA–Insulin Pathways

The study of metabolic stress conditions on islets was reproduced on the INS-1E cell line, the most widely studied beta-cell model derived from a rat insulinoma. Surprisingly, high glucose exposure for 48 h was the only condition inducing GSIS failure in INS-1E cells, as GSIS was increased by glucolipotoxicity by 72% and showed a tendency to increase during lipotoxicity (Figure 2a). However, INS-1E beta cells showed a 50% decrease in IC after glucolipotoxicity and glucotoxicity but not lipotoxicity (*p* < 0.001) (Figure 2b) in total agreement with the alterations in human islets (Figure 1b). In addition, we measured cell growth and oxygen consumption rates (OCR) as a marker of the ATP regeneration capacity of the cells. The results showed that glucotoxicity alone reduced cell numbers by 13% after 72 h of culture (*p* < 0.05), whereas cell numbers were reduced by 32% and 38% by glucolipo- and lipotoxicity alone after 24 h (*p* < 0.001 and *p* < 0.0001 respectively) (Appendix A). OCR was also measured between control and glucolipotoxicity in permeabilized cells directly supplied with pyruvate and ADP. It revealed a clear decrease in oxygen consumption of 85% (*p* < 0.01) after glucolipotoxicity treatment (Figure 2c and Appendix A). Taken together, these data indicate that glucolipotoxicity influences cell viability and may have an impact on the ability of mitochondria to use pyruvate.

As with human islets, Appendix A shows the absolute concentration of metabolites in the medium before INS-1E incubation and after 48 h of incubation, enabling the rate of metabolite consumption or secretion to be determined in nmol per million cells per 48 h. Glutamine, glucose, and arginine are the substrates most consumed by INS-1E cells while catabolic products such as glutamate, alanine, and lactate are the most secreted. As with human islet analyses, a correlogram was used to identify metabolites correlated with both INS-1E GSIS and IC treated with control, gluco-, lipo-, and glucolipotoxicity. However, as the loss of GSIS and IC did not correlate with the response to glucolipotoxicity (Figure 2a,b), the identification of metabolites with significant variation shared between GSIS and IC was irrelevant, unlike in human islets (Figure 2d). In INS-1E cells, we identified 9/27 significantly altered metabolites in the culture supernatant that were linked to altered IC, while a further 12/27 metabolites were linked to altered GSIS. Altogether, 21/27 metabolites were modified according to loss of function of INS-1E.

As in human islets, pyruvate secretion by INS-1E cells under glucolipo- and glucotoxicity increased 2.5-fold after 48 h (2.07 µmol, and 2.37 µmol respectively, vs. 0.86 µmol in control conditions) but no change after lipotoxicity (Figure 2e; Appendix A *p* < 0.001). On the other hand, citrate secretion is decreased by almost 25% in cases of gluco-, lipo-, and glucolipotoxicity (0.16 µmol, 0.14 µmol and 0.12 µmol respectively, vs. 0.2 µmol in control conditions) (Figure 2f; Appendix A *p* < 0.001).

Concurrent analyses of the INS-1E exometabolome with the traditional endometabolome under glucolipo-, gluco-, and lipotoxicity showed citrate accumulation in cells of the order of 0.08 mM +/− 0.009 (Appendix A), which cannot explain the 0.1 µmol/10^6^ cells/24 h reduction observed in glucolipotoxicity (Appendix A; Appendix A). Glycerol accumulation increased from 0.03 to 0.102 mM/10^6^ cells (Appendix A) as did triglycerides (from 10 to 68 µmol/10^6^ cells) during glucolipotoxicity (Appendix A). Triglyceride secretion was evident with a secretion from 6 to 132 µmol/10^6^ cells/24 h during glucolipotoxicity (Figure 2g). We interpret that citrate metabolism may depend on lipid synthesis. Indeed, some of the citrate produced by the mitochondria can be relocated in the cytoplasm and feed the de novo lipogenesis pathway (Figure 2h). In addition, for the INS-1E model, several de novo lipogenesis enzymes were monitored and quantified: fatty acid synthase (FAS), acetyl-CoA carboxylase (ACC), and ATP citrate lyase (ACLY). Glucolipotoxicity increased ACLY and FAS quantities more than twofold (*p* < 0.01 and *p* < 0.05, respectively) but did not significantly affect ACC. Lipotoxicity alone did not affect the relative quantity of lipogenic enzymes in INS-1E after 48 h. Glucotoxicity alone increased the relative quantities of ACLY, ACC, and FAS (Figure 2i).

Then, we profiled mitochondrial dehydrogenase expression as in human islets. IDH was not downregulated; we observed a clear reduction in OGDH quantity in glucolipotoxicity (39%, *p* < 0.05) and glucotoxicity (44%, *p* < 0.05) conditions. Lipotoxicity did not affect dehydrogenase levels but pPDH was clearly decreased with a 42% reduced pPDH/PDH ratio, a feature not observed in human islets (Figure 2j).

## 4. Discussion

In this study, we performed a comprehensive characterization of the metabolites secreted and consumed by cultured human pancreatic islets under various conditions, using NMR metabolomic approaches. These analytical methods are perfectly suited to determine the biological activity alteration and evaluate the positive and negative effects of nutrients in islets but also in a variety of cell types and models [33]. Two major constraints in our study were the limited quantity of human pancreatic islets and the timing of mimicking chronic, not acute, nutrient-induced alteration for ex vivo analyses. Of note, the study was designed for 48 h of metabolic stress condition, as this duration was chosen to be both minimal and sufficient to induce detectable modifications of islet physiology, as previously described [22]. Thus, we found clear discrimination between islets from healthy donors and those from donors with T2D. Islets from T2D donors consumed more glucose and secreted greater amounts of pyruvate and citrate than islets from healthy donors. These results indicate that the glycolysis to TCA axis was significantly dampened in the context of T2D. Similar changes were observed in healthy donors and INS-1E cells exposed to chronic high glucose. Our results demonstrated that glycolysis was upregulated in diabetic islets, which show a clear increase in glucose consumption and pyruvate and lactate secretion. These findings are in agreement with the upregulation of selected glycolytic enzymes observed in rat islets with T2D [34] and in rat islets and cell lines cultured in the presence of high glucose concentration [18,19,35,36].

We also demonstrate that glucolipotoxic conditions induced the downregulation of mitochondrial OGDH in islets from healthy donors and in INS-1E cells. This regulation should be associated with a reduction in TCA levels, ATP synthesis, and insulin secretion. In contrast to previous reports in rodent islets based on combined transcriptomic–proteomic [18] or combined proteomic–phosphoproteomic analyses [19], IDH2 expression was not downregulated in our study and the change in the pPDH/PDH ratio observed by both research groups was demonstrated only in INS-1E cells and not in human islets. These discrepancies are probably related to the well-described disparities between human and rat models, which have long been reported in the literature [32] and have recently been reviewed in the context of metabolomic studies [20].

Our study of INS-1E cells reveals that both mitochondrial oxygen consumption and coupling efficiency are impaired. These changes may be associated with the inability of increased intracellular glucose to raise cytosolic ATP levels and drive glucose-stimulated insulin release. Contrary to islets secreting citrate and pyruvate, we observed a marked increase in triglyceride secretion by INS-1E cells in the presence of high glucose concentration, and overexpression of the lipogenic enzymes ACLY, ACC, and FAS. An increase in triglyceride biosynthesis was also observed in islets isolated from obese mouse models of diabetes [37]. The low activity of ACLY, the first enzyme involved in the de novo lipogenic pathway, in human islets (24 nmol product/min/mg protein) compared with rat islets (96 nmol product/min/mg protein) [32] could explain the functional discrepancies between human and rodent models. Overall, these data suggest that high glucose alone may be sufficient to induce functional changes in rat INS-1E cells, and this raises the question of whether rodent islets, but not human islets, would be able to activate the de novo lipogenesis pathway [20].

Altogether, our NMR-based metabolomic results from pancreatic islets and INS-1E cells support a model in which higher glucose concentrations trigger functional alteration of beta cells, which is associated with GSIS and IC failure and the profound adaptation of cellular and mitochondrial metabolism (Figure 3). The excess of energy provided by the increase in intracellular glucose is processed either toward pyruvate and citrate secretion (in the case of islets) or pyruvate and triglyceride (in the case of INS-1E cells). This model provides integrated support compatible with our previous work highlighting the role of mitochondria–ER contact and mitochondrial dehydrogenases in the progression of diabetic disease [22,38] (Appendix A).

## Figures and Tables

**Figure 1 nutrients-15-04791-f001:**
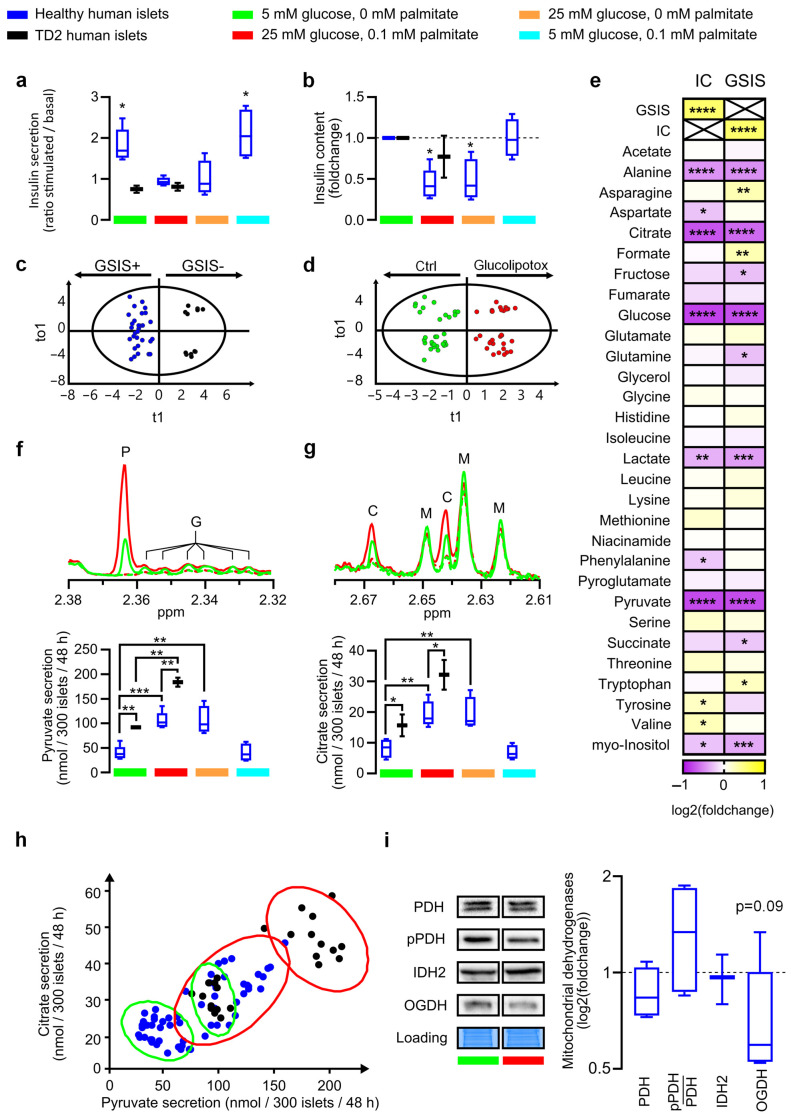
Functional analysis of human pancreatic islets and metabolite production and secretion analyses under glucolipotoxicity: (**a**,**b**) Boxplots showing respectively stimulated/basal insulin secretion and corresponding cellular Insulin Content (IC) of human pancreatic islets during GSIS after 48 h treatment with indicated glucose and palmitate concentration (*n* = 5 healthy human islets D1–D5, and *n* = 2 T2D islets D10–D11). Significance in (**a**) represents the difference between stimulated vs. basal insulin secretion, and significance in (**b**) represents the difference between control and treated conditions. (**c**,**d**) 0-PLS-DA score plot derived from 1H NMR metabolite quantification discriminating human GSIS positive islets vs. non-secreting ones after 48 h in control medium (**c**) and after 48 h in control vs. glucolipotoxicity medium (**d**). The major metabolites contributing to the cluster separation were evidenced by the derived models’ VIP values detailed in Appendix A. (**e**) Log2 fold change in the correlation coefficient between islet insulin secretion after glucose stimulation (GSIS) or insulin content (IC) and metabolite concentration in medium after 48 h in all treatments (control, gluco-, lipo-, or glucolipotoxicity) of D1–D5 islet donors. Significance represents the significativity of the calculated correlation. (**f**,**g**) Representative high-resolution NMR spectra (**upper panel**) and corresponding metabolite quantification (**lower panel**). P, G, C, and M 1H-NMR peaks correspond to specific peaks of pyruvate, glutamate, citrate, and methionine, respectively. Green lines represent control culture conditions compared to high-glucose (25 mM) and high-palmitate (0.1 mM) culture conditions in red. Dotted line and full line represent medium at 0 and 48 h culture, respectively. (**f**) Pyruvate and (**g**) citrate secretions of 300 pancreatic islets after 48 h of incubation with indicated culture conditions. (**h**) Scatter plot of pancreatic islet pyruvate and citrate secretions after 48 h in control culture condition (green ellipse) and in high-glucose (25 mM) and palmitate (0.1 mM) culture condition (red ellipse) in healthy islets (blue) and DT2 islets (black). Ellipse represents a confidence interval of 95%. (**i**) Representative Western blots and quantitative analyses of key mitochondrial dehydrogenases namely PDH, pPDH, IDH2, OGDH in human pancreatic islets of 5 donors (D1, D4–D7) cultured 48 h under control or glucolipotoxicity. Student *t*-test was performed to compare enzyme expression between control and glucolipotoxicity conditions. Error bars represent standard deviation, * *p* < 0.05, ** *p* < 0.01, *** *p* < 0.001, **** *p* < 0.0001.

**Figure 2 nutrients-15-04791-f002:**
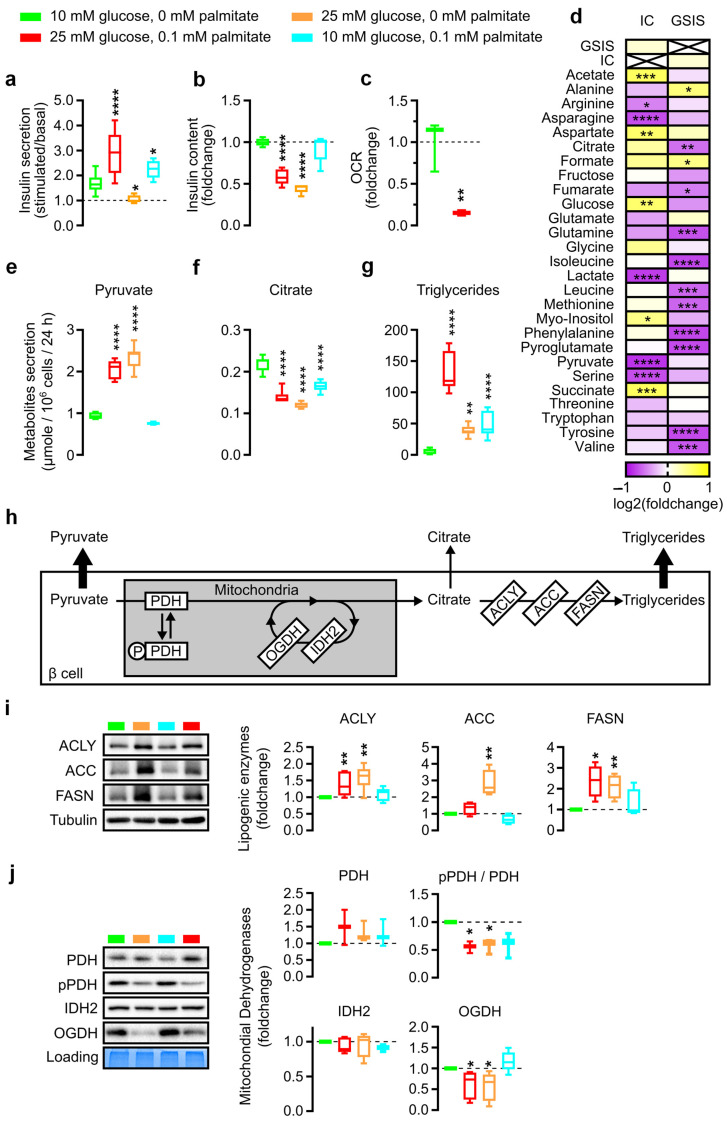
Functional analysis of INS-1E beta cell and metabolite production analyses under glucolipotoxicity. (**a**,**b**) Boxplots showing respectively Stimulated/Basal insulin secretion and corresponding cellular insulin content (IC) of INS-1E cells during GSIS after 48 h treatment with indicated glucose and palmitate concentration. Significance in (**a**) represents the difference between Stimulated vs. Basal insulin secretion and significance in (**b**) represents the difference between control and treated conditions. (**c**) Boxplots showing oxygen consumption rate (OCR) of INS-1E permeabilized cells after pyruvate and ADP addition following 48 h treatment in corresponding glucose and palmitate concentration. (**d**) Log2 fold change in the correlation coefficient between islet insulin secretion after glucose stimulation (GSIS) or insulin content (IC) and metabolite concentration in medium after 48 h in all treatments (control, gluco-, lipo-, or glucolipotoxicity) of INS-1E cells. Significance represents the significativity of the calculated correlation. (**e**–**g**) Pyruvate (**e**), citrate (**f**), and triglycerides (**g**) secretion normalized by cell number after 48 h incubation with indicated culture conditions. (**h**) Pyruvate fate in oxidative phosphorylation metabolism and de novo lipogenesis. (**i**,**j**) Representative Western blot and quantitative analyses of ACLY, ACC, FASN (**i**) and PDH, pPDH, IDH2, OGDH (**j**) in INS-1E cells. Error bars represent standard deviation. * *p* < 0.05, ** *p* < 0.01, *** *p* < 0.001, **** *p* < 0.0001.

**Figure 3 nutrients-15-04791-f003:**
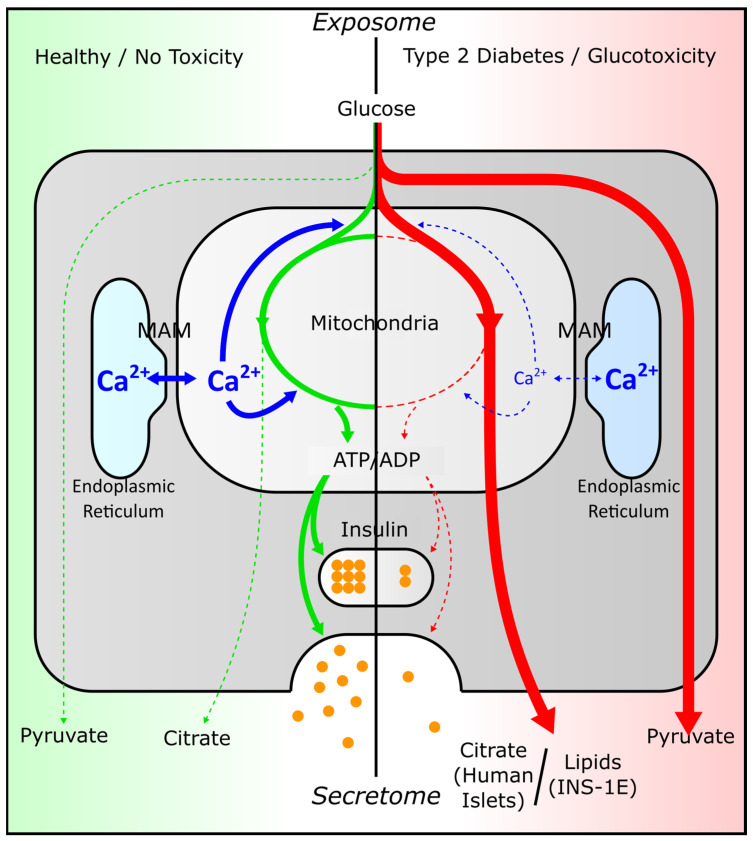
Model presenting how glucolipotoxicity could trigger a mitochondrial bypass that leads to metabolite excretion rather than energy production.

**Table 1 nutrients-15-04791-t001:** Properties of each batch of human islets regarding donor physiopathology and death.

Donor	D1	D2	D3	D4	D5	D6	D7	D8	D9	D10	D11	Mean	SEM
identifier	HP 1907	HP 1905	HP 1908	HO 2001	HP 1910	HI-28	HI-29	HI-20	HM 88	HM 84	HM 92		
Origin of islets	CEED	CEED	CEED	CEED	CEED	ECIT	ECIT	ECIT	LCTD	LCTD	LCTD		
isolation centre	Strasbourg	Strasbourg	Strasbourg	Strasbourg	Strasbourg	Geneva	Geneva	Geneva	Montpellier	Montpellier	Montpellier		
BMI (kg/m^2^)	25.9	19.1	20.5	26.7	27.8	21.9	25.6	27.2	26	30.5	30.1	25.6	3.7
age (years)	58	60	86	64	61	55	59	46	26	62	68	58.6	14.6
Sex (M/F)	M	M	F	F	M	F	M	M	M	M	M	27% F/73% M	
plasma glucose test	<11.1 mmol/L	<11.1 mmol/L	ND	ND	<11.1 mmol/L	<11.1 mmol/L	<11.1 mmol/L	<11.1 mmol/L	ND	<11.1 mmol/L	ND		
cause of death	Stroke	Stroke	subdural hematoma	Stroke	Stroke	no death (auto-transplantation)	Stroke	Stroke	Suicide	Trauma	Trauma		
Warm ischemia time	54 min	2 min 30	35 min	<2 h	<2 h	<2 h	<2 h	62 min	<2 h	<2 h	<2 h		
Donor history of diabete?	No	No	No	No	No	No	No	No	No	T2D	T2D		
therapy at time of death										Metformin	Metformin		

## Data Availability

The newly generated 1H-NMR data have been deposited in the MetaboLights database with the following accession number MTBLS3541.

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
