# Peer review of "Human Pancreatic Islets React to Glucolipotoxicity by Secreting Pyruvate and Citrate"

_nutrients, 2023, doi:10.3390/nu15224791_

Round 1

Reviewer 1 Report

Comments and Suggestions for Authors

Main problem is the data presentation, inconsistency or lack of explanation why different data sets are used in different analysis, repeated lack of explanation of abbrevations. Attached are my comments.

Reviewer 2 Report

Comments and Suggestions for Authors

The authors investigated the effect of glucolipotoxicity on cellular metabolite in human pancreatic beta cells and INS-1 cells. And they also demonstrated that glucolipotoxicity induced the increase of pyruvate/citrate/lipid secretion through the downregulation of mitochondrial OGDM while it induced the decrease of insulin secretion/contents.

However, additional experiments are required:

1) effects of OGDM overexpression on insulin, pyruvate, citrate or lipid secretion in high glucose and/or palmitate treated INS-1 cells.

2) effects of OGDM overexpression on the expression levels of PDH, pPDH, IDH2, ACLY, ACC or FASN in high glucose and/or palmitate treated INS-1 cells.

3) effects of OGDM siRNA (knock-down) on insulin, pyruvate, citrate or lipid secretion in normal condition of INS-1 cells.

4) Please change tables 2 and 3 to supplementary data. Instead, please make a new figures of the main data in tables 2 and 3 (e.g. pyruvate, citrate, etc.)

Comments on the Quality of English Language

Minor editing of English language required

Author Response

The authors would like to express their gratitude to the reviewer for the valuable suggestions. Our study delves into the exometabolome, which offers a wealth of insights into the nutritional environment and the secreted metabolites by cells, shedding light on the modified intracellular mechanisms within human pancreatic islets under glucolipotoxicity conditions. Our research has led to the formulation of several hypotheses aimed at explaining these alterations.
We acknowledge the reviewer's suggestion of employing gene overexpression and silencing techniques in order to obtain a more comprehensive characterization of the intracellular physiopathological mechanisms. Indeed, such an approach would provide a significantly deeper understanding of the intricate processes at play. However, undertaking such a study would extend beyond the scope of our current project. Moreover, our team's expertise may not be fully equipped to tackle the complexities associated with OGDHM, a multiprotein complex that demands advanced strategies for effective silencing. Nevertheless, we appreciate the suggestion and will consider it for future investigations or collaborations.

The title of the article has been updated to "Human Pancreatic Islets Respond to Glucolipotoxicity by Secreting Pyruvate and Citrate" to better encapsulate the main focus of your study and align with the reviewers' recommendations for improved presentation. 
We appreciate the reviewer's feedback, and have relocated Tables 2 and 3 to the supplementary data. This adjustment ensures that these tables remain accessible to interested readers without burdening the core study.
The emphasis on citrate and pyruvate is now prominently showcased in Figures 1F, 1G, and 1H, providing a more detailed and focused visualization of our findings. Additionally, we have included a summary view of the metabolic secretion changes in Figure 1E, which serves as a comprehensive overview of the alterations observed. We believe that these changes will enhance the clarity and readability of our research, and we thank the reviewer for this constructive input in helping us improve our work.

Reviewer 3 Report

Comments and Suggestions for Authors

Perrier and colleagues present an interesting study exploring the impact of glucolipotoxic conditions on pancreatic beta-cell metabolism. Whilst others have published in this area, typically using rodent cell lines; this study additionally explores this question in human islets exposed to these conditions, and also in T2D islets. This significantly improves the importance of these findings. The experimental approach was sound (aside from the comment below), and the conclusions drawn reasonable. The final summary figure was helpful.

The authors suggest that the glucolipotoxic conditions cause a reduction in cell viability in the INS cells. No viability proxy was assessed in human islets, but it is likely that there would be some loss of cells here also. Given that the supernatant in which these cells have been cultured for 48h is used for metabolomic analysis, can the authors be sure that the increased levels of metabolites observed are not simply due to the increased number of lysed cells in these samples?

Minor issues:

Pg 4: Please include antibody dilutions for all antibodies used.

Tables: The metabolic information provided in the tables in this study is somewhat overwhelming. I would suggest presenting the information in this format in the supplementary tables, but then to find a more reader friendly way to summarise these data within the main figures.

P16 line 395. The authors state that cell numbers were reduced by glucolipotoxic conditions. It was not made clear whether any measure of cell viability was collected to inform these conclusions (e.g. trypan blue). Differences in cell number between conditions may not reflect changes in viability.

Sup Fig 6c: Can the authors please include information on the number of independent replicates conducted to generate these data.

Reviewer 4 Report

Comments and Suggestions for Authors

Perrier et al. examined the impact of glucolipotoxicity on pancreatic islets, with a specific emphasis on mitochondria function. The study highlights a novel metabolomics method and provides some new information on the metabolic alterations of beta cells in response to the diet-induced stress condition. Although the conclusion is confirmatory, it provides some new evidence. I commend the authors for sophisticated statistical analysis of their profiling results and a well-written manuscript.

I have some minor points to improve the manuscript:

  1. The authors should consider replacing the title.
  2. Line 85 typo “inc”.
  3. Line 625 typo “TD2”.
  4. I felt like these tables were hard to read. It might be better to considering add up or down arrow indicators to the table to help the readers.

Round 2

Reviewer 2 Report

Comments and Suggestions for Authors

The authors investigated the effect of glucolipotoxicity on cellular metabolite in human pancreatic beta cells and INS-1 cells. And they also demonstrated that glucolipotoxicity induced the increase of pyruvate/citrate/lipid secretion through the downregulation of mitochondrial OGDM while it induced the decrease of insulin secretion/contents. Although this study lacks mechanistic studies on the role of OGDM in pancreatic beta cells, this study focuses on the analyses of exometabolome using human pancreatic islets. And the manuscript was revised to the reviewer's suggestion.

Author Response

The authors appreciate that the corrections made to the manuscript satisfy the reviewer. Thank you very much for your consideration
